# Ada-NETS: Face Clustering via Adaptive Neighbour Discovery in the Structure Space

**Yaohua Wang**[*], **Yaobin Zhang** [*], **Fangyi Zhang, Ming Lin, YuQi Zhang**
Alibaba Group
{xiachen.wyh, zhangyaobin.zyb, zhiyuan.zfy, ming.l,
gongyou.zyq}@alibaba-inc.com

**Senzhang Wang**[†]
Central South University
szwang@csu.edu.cn

**Xiuyu Sun**[†]
Alibaba Group
xiuyu.sxy@alibaba-inc.com

## Abstract

Face clustering has attracted rising research interest recently to take advantage of massive amounts of face images on the web. State-of-the-art performance has been achieved by Graph Convolutional Networks (GCN) due to their powerful representation capacity. However, existing GCN-based methods build face graphs mainly according to $k$NN relations in the feature space, which may lead to a lot of noise edges connecting two faces of different classes. The face features will be polluted when messages pass along these noise edges, thus degrading the performance of GCNs. In this paper, a novel algorithm named Ada-NETS is proposed to cluster faces by constructing clean graphs for GCNs. In Ada-NETS, each face is transformed to a new structure space, obtaining robust features by considering face features of the neighbour images. Then, an adaptive neighbour discovery strategy is proposed to determine a proper number of edges connecting to each face image. It significantly reduces the noise edges while maintaining the good ones to build a graph with clean yet rich edges for GCNs to cluster faces. Experiments on multiple public clustering datasets show that Ada-NETS significantly outperforms current state-of-the-art methods, proving its superiority and generalization. Code is available at https://github.com/damo-cv/Ada-NETS.

## 1 Introduction

The number of images on the web increases rapidly in recent years, a large portion of which are human-centred photos. Understanding and managing these photos with little human involvement are demanding, such as associating together photos from a certain person. A fundamental problem towards these demands is face clustering (Driver & Kroeber, 1932).

Face clustering has been thoroughly investigated in recent years. Significant performance improvements (Wang et al., 2019b; Yang et al., 2019; 2020; Guo et al., 2020; Shen et al., 2021) have been obtained with Graph Convolutional Networks due to their powerful feature propagation capacity. The representative DA-Net (Guo et al., 2020) and STAR-FC (Shen et al., 2021) use GCNs to learn enhanced feature embedding by vertices or edges classification tasks to assist clustering.

However, the main problem restricting the power of existing GCN-based face clustering algorithms is the existence of **noise edges** in the face graphs. As shown in Figure 1 (b), a noise edge means the connection between two faces of different classes. Unlike common graph datasets such as Citeseer, Cora and Pubmed with explicit link relation as edges (Kipf & Welling, 2017), face images do not contain explicit structural information, but only deep features extracted from a trained CNN model. Therefore, face images are treated as *vertices*, and the *edges* between face images are usually constructed based on the $k$NN (Cover & Hart, 1967) relations when building the graph: Each face serves

---

[*]Equal contribution
[†]Corresponding author

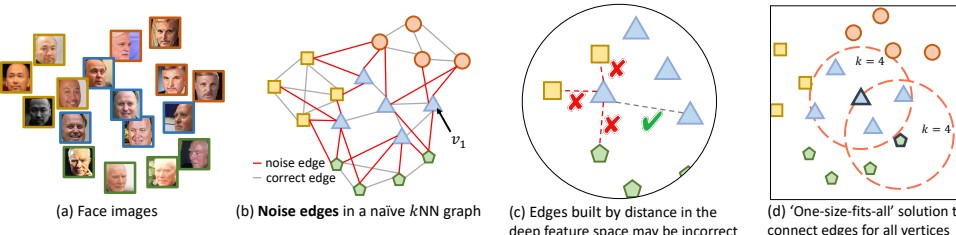

(a) Face images   (b) **Noise edges** in a naïve $k$NN graph   (c) Edges built by distance in the deep feature space may be incorrect   (d) 'One-size-fits-all' solution to connect edges for all vertices

Figure 1: The noise edges problem in GCN-based face clustering. Different shapes in figures represent different classes. (a) Face images to be clustered. (b) Noise edges are introduced when constructing graphs based on naïve $k$NN. (c) Connecting edges by feature distance may lead to noise edges. (d) The existing "One-size-fits-all" solution using a fixed number of neighbours for each vertex introduces many noise edges.

as a probe to retrieve its $k$ nearest neighbours by deep features (Wang et al., 2019b; Yang et al., 2019; 2020; Guo et al., 2020; Shen et al., 2021). The $k$NN relations are not always reliable because deep features are not accurate enough. So the noise edges are introduced to the graph along with the $k$NN. The noise edges problem is common in face clustering but has received little research attention. For example, the graph used in (Yang et al., 2020; Shen et al., 2021) contains about $38.23\%$ noise edges in testing. The noise edges will propagate noisy information between vertices, hurting their features when aggregation, and thus resulting in inferior performance. In Figure 1 (b), the triangular vertex $\mathbf{v}_1$ is connected with three vertices of different classes, and will it be polluted by them with messages passing in the graph. Therefore, GCN-based linkage prediction cannot effectively address the noise edges problem in related works (Wang et al., 2019b; Yang et al., 2020; Shen et al., 2021).

The challenges of removing the noise edges in the face graphs are two-fold as shown in Figure 1 (c) (d). First, the representation ability of deep features is limited in real-world data. It is difficult to tell whether two vertices are of the same class solely according to their deep features, thus noise edges are inevitably brought by connecting two vertices of different classes. Second, it is hard to determine how many edges to connect for each vertex when building a graph: Too few edges connected will result in insufficient information aggregation in the graph. Too many edges connected will increase the number of noise edges, and the vertex feature will be polluted by wrongly connected vertices. Although Clusformer (Nguyen et al., 2021) and GAT (Velickovic et al., 2018) try to reduce the impact of the noise edges by the attention mechanism, the connections between various vertices are very complex, and thus it is difficult to find common patterns for the attention weight learning (Yang et al., 2020).

To overcome these tough challenges, the features around each vertex are taken into account because they can provide more information. Specifically, each vertex feature representation can be improved when considering other vertices nearby. This is beneficial to address the representation challenge in Figure 1 (c). Then, the number of edges between one vertex and others can be learned from feature patterns around it instead of a manually designed parameter for all vertices. This learning method can effectively reduce the connection of noise edges which is crucial to address the second challenge in Figure 1 (d). Based on the ideas above, a novel clustering algorithm, named **Adaptive Neighbour discovEry in the strucTure Space** (Ada-NETS), is proposed to handle the noise edges problem for clustering. In Ada-NETS, a *structure space* is first presented in which vertices can obtain robust features by encoding more texture information after perceiving the data distribution. Then, a *candidate neighbours quality criterion* is carefully designed to guide building less-noisy yet rich edges, along with a learnable *adaptive filter* to learn this criterion. In this way, the neighbours of each vertex are adaptively discovered to build the graph with clean and rich edges. Finally, GCNs take this graph as input to cluster human faces.

The main contributions of this paper are summarized as follows:

- To the best of our knowledge, this is the first paper to address the noise edges problem when building a graph for GCNs on face images. Simultaneously, this paper demonstrates its causes, great impact, weaknesses in existing solutions and the challenges to solve it.

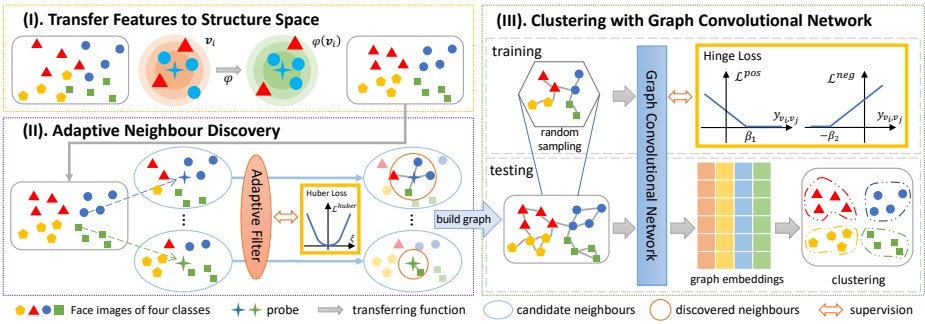

Figure 2: The framework of Ada-NETS. (I). The features are transformed to the structure space to obtain better similarity metrics. (II). The neighbours of each vertex are discovered by an adaptive filter. (III). A graph is built with the neighbour relations discovered by (II) and the graph is used by the GCN model to classify vertex pairs. The final clustering results are obtained using embeddings from GCNs to link vertex pairs with high similarities.

- The proposed Ada-NETS can alleviate the noise edges problem when building a graph on face images, thus improve GCNs greatly to boost clustering performance.
- State-of-the-art performances are achieved on clustering tasks by Ada-NETS, surpassing the previous ones by a large margin on the face, person, and clothes datasets.

## 2 RELATED WORK

**Face Clustering**  Face clustering tasks often face large-scale samples and complex data distribution, so it has attracted special research attention. Classic unsupervised methods are slow and cannot achieve good performances for their naive distribution assumptions, such as convex-shaped data in K-Means (Lloyd, 1982) and similar density of data in DBSCAN (Ester et al., 1996)). In recent years, GCN-based supervised methods are proved to be effective and efficient for face clustering. L-GCN (Wang et al., 2019b) deploys a GCN for linkage prediction on subgraphs. DS-GCN (Yang et al., 2019) and VE-GCN (Yang et al., 2020) both propose two-stage GCNs for clustering based on the big $k$NN graph. DA-Net (Guo et al., 2020) conducts clustering by leveraging non-local context information through density-based graph. Clusformer (Nguyen et al., 2021) clusters faces with a transformer. STAR-FC (Shen et al., 2021) develops a structure-preserved sampling strategy to train the edge classification GCN. These achievements show the power of GCNs in representation and clustering. However, the existing methods mostly build face graphs based on $k$NN, which contain a lot of noise edges. When building these graphs, the similarities between vertices are obtained only according to deep features which are not always accurate, and the number of edges for each vertex is fixed or determined by a similarity threshold.

**Graph Convolutional Networks**  GCNs are proposed to deal with non-Euclidean data and have shown their power in learning graph patterns. It is originally used for transductive semi-supervised learning (Kipf & Welling, 2017) and is extended to inductive tasks by GraphSAGE (Hamilton et al., 2017) that learns the feature aggregation principle. To further expand the representation power of GCNs, learnable edge weights are introduced to the graph aggregation in the Graph Attention Network (GAT) (Velickovic et al., 2018). In addition to face clustering, GCNs are also used in many tasks such as skeleton-based action recognition (Yan et al., 2018), knowledge graph (Schlichtkrull et al., 2018) and recommend system (Ying et al., 2018). However, these methods are proposed on structure data, where the edges are explicitly given. The GCNs may not perform well on face image datasets if the graph is constructed with a lot of noise edges.

## 3 METHODOLOGY

Face clustering aims to divide a set of face samples into groups so that samples in a group belong to one identity, and any two samples in different groups belong to different identities. Given a set of feature vectors $\mathcal{V} = \{\mathbf{v}_1, \mathbf{v}_2, ..., \mathbf{v}_i, ..., \mathbf{v}_N \mid \mathbf{v}_i \in \mathbb{R}^D\}$ extracted from face images by a trained

Figure 3: Adaptive neighbour discovery process. $\mathbf{k}^{\text{off}}$ is the extreme point of $Q(j)$. In training phase, adaptive filter learns to fit $\mathbf{k}^{\text{off}}$. In testing phase, adaptive filter estimates $\mathbf{k}^{\text{off}}$ and removes the candidate neighbours with orders beyond the predicted $\hat{\mathbf{k}}^{\text{off}}$.

CNN model, the clustering task assigns a group label for each vector $\mathbf{v}_i$. $N$ is the total number of samples, and $D$ is the dimension of each feature. The Ada-NETS algorithm is proposed as shown in Figure 2 to cluster faces by dealing with the noise edges in face graphs. Firstly, the features are transformed to the proposed structure space for an accurate similarity metric. Then an adaptive neighbour discovery strategy is used to find neighbours for each vertex. Based on the discovery results, a graph with clean and rich edges is built as the input graph of GCNs for the final clustering.

## 3.1 STRUCTURE SPACE

The noise edges problem will lead to the pollution of vertices features, degrading the performance of GCN-based clustering. It is difficult to determine whether two vertices belong to the same class just based on their deep features, because two vertices of different classes can also have a high similarity, thus introducing noise edges. Unfortunately, to the best of our knowledge, almost all existing methods (Wang et al., 2019b; Yang et al., 2020; Guo et al., 2020; Shen et al., 2021) build graphs only based on the pairwise cosine similarity between vertices using deep features. In fact, the similarity metric can be improved by considering the structure information, that is the neighbour-hood relationships between images of the dataset. Based on this idea, the concept of the structure space is proposed to tackle this challenge. In the structure space, the features can encode more texture information by perceiving the data distribution thus being more robust (Zhang et al., 2020). A transforming function $\varphi$ is deployed to convert one feature $\mathbf{v}_i$ into the structure space, noted as $\mathbf{v}_i^s$:

$$\mathbf{v}_i^s = \varphi\left(\mathbf{v}_i | \mathcal{V}\right), \forall i \in \{1, 2, \cdots, N\}. \tag{1}$$

As shown in Figure 2 (I), with the help of the structure space, for one vertex $\mathbf{v}_i$, its similarities with other vertices are computed by the following steps: First, $k$NN of $\mathbf{v}_i$ are obtained via an Approximate Nearest-neighbour (ANN) algorithm based on the cosine similarity between $\mathbf{v}_i$ and the other vertices, noted as $\mathcal{N}(\mathbf{v}_i, k) = \{\mathbf{v}_{i_1}, \mathbf{v}_{i_2}, \cdots, \mathbf{v}_{i_k}\}$. Second, motivate by the kernel method (Shawe-Taylor & Cristianini, 2004), instead of directly solving the form of $\varphi$, we define the similarity of $\mathbf{v}_i$ to each of its candidates in the structure space by

$$\begin{aligned}\kappa\left(\mathbf{v}_i, \mathbf{v}_{i_j}\right) &= \left\langle \mathbf{v}_i^s, \mathbf{v}_{i_j}^s \right\rangle \\ &\triangleq (1 - \eta) s^{\text{Jac}}\left(\mathbf{v}_i, \mathbf{v}_{i_j}\right) + \eta s^{\text{cos}}\left(\mathbf{v}_i, \mathbf{v}_{i_j}\right), \quad \forall j \in \{1, 2, \cdots, k\},\end{aligned} \tag{2}$$

where $\eta$ weights the cosine similarity $s^{\text{cos}}\left(\mathbf{v}_i, \mathbf{v}_{i_j}\right) = \frac{\mathbf{v}_i \cdot \mathbf{v}_{i_j}}{\|\mathbf{v}_i\| \|\mathbf{v}_{i_j}\|}$ and the Jaccard similarity $s^{\text{Jac}}\left(\mathbf{v}_i, \mathbf{v}_{i_j}\right)$ that is inspired by the common-neighbour-based metric (Zhong et al., 2017). With the definitions above, $\kappa\left(\mathbf{v}_i, \mathbf{v}_{i_j}\right)$ measures the similarity between $\mathbf{v}_i$ and $\mathbf{v}_{i_j}$ in the structure space.

## 3.2 ADAPTIVE NEIGHBOUR DISCOVERY

The existing methods link edges from naive $k$NN relations retrieved by deep features (Wang et al., 2019b; Yang et al., 2020; Shen et al., 2021) or using a fixed similarity threshold (Guo et al., 2020). These methods are all one-size-fits-all solutions and the hyper-parameters have a great impact on the performance. To address this issue, the adaptive neighbour discovery module is proposed to learn from the features pattern around each vertex as shown in Figure 2 (II).

For the vertex $\mathbf{v}_i$, its *candidate neighbours* of size $j$ are the $j$ nearest neighbour vertices based on the similarity of their deep features, where $j = 1, 2, \cdots, k$. Its *neighbours* mean one specific sized candidate neighbours that satisfy some particular criterion described as follows. The edges between $v_i$ and all its neighbours are constructed.

### 3.2.1 CANDIDATE NEIGHBOURS QUALITY CRITERION

Motivated by the vertex confidence estimation method (Yang et al., 2020), a heuristic criterion is designed to assess the quality of candidate neighbours for each probe vertex. Good neighbours should be clean, i.e., most neighbours should have the same class label with the probe vertex, so that noise edges will not be included in large numbers when building a graph. The neighbours should also be rich, so that the message can fully pass in the graph. To satisfy the two principles, the criterion is proposed according to the $F_{\beta}$-score (Rijsbergen, 1979) in information retrieval. Similar to visual grammars (Nguyen et al., 2021), all candidate neighbours are ordered by the similarity with the probe vertex in a sequence. Given candidate neighbours of size $j$ probed by vertex $\mathbf{v}_i$, its quality criterion $Q(j)$ is defined as:

$$Q(j) = F_{\beta}^j = (1 + \beta^2) \frac{Pr^j Rc^j}{\beta^2 Pr^j + Rc^j}, \tag{3}$$

where $Pr^j$ and $Rc^j$ are the precision and recall of the first $j$ candidate neighbours with respect to the label of $\mathbf{v}_i$. $\beta$ is a weight balancing precision and recall. A higher $Q$-value indicates a better candidate neighbours quality.

### 3.2.2 ADAPTIVE FILTER

With the criterion above, $\mathbf{k}^{\text{off}}$ is defined as the heuristic ground truth value of the number of neighbours to choose:

$$\mathbf{k}^{\text{off}} = \arg \max_{j \in \{1,2,...k\}} Q(j). \tag{4}$$

As shown in Figure 3, the adaptive filter estimates $\mathbf{k}^{\text{off}}$ by finding the position of the highest $Q$-value on the $Q$-value curve. The input of the adaptive filter is the feature vectors $[\mathbf{v}_i, \mathbf{v}_{i_1}, \mathbf{v}_{i_2}, \cdots, \mathbf{v}_{i_k}]^T \in \mathbb{R}^{(k+1) \times D}$. In training, given a mini-batch with $B$ sequences, adaptive filter is trained using the Huber loss (Huber, 1992):

$$\mathcal{L}^{\text{Huber}} = \frac{1}{B} \sum_{b=1}^{B} \mathcal{L}_b^{\text{Huber}},$$

$$\text{where } \mathcal{L}_b^{\text{Huber}} = \left\{ \begin{array}{ll} \frac{1}{2}\xi^2, & \xi < \delta, \\ \delta\xi - \frac{1}{2}\delta^2, & \text{otherwise,} \end{array} \right. \qquad \xi = \frac{|\hat{\mathbf{k}}_b^{\text{off}} - \mathbf{k}_b^{\text{off}}|}{\mathbf{k}_b^{\text{off}}}, \tag{5}$$

$\hat{\mathbf{k}}_b^{\text{off}}$ is the prediction of $\mathbf{k}_b^{\text{off}}$, and $\delta$ is an outlier threshold. With each prediction $\hat{\mathbf{k}}^{\text{off}}$ from $\hat{\mathbf{k}}_b^{\text{off}}$, the candidate neighbours with orders beyond $\hat{\mathbf{k}}^{\text{off}}$ are removed and the left will be treated as neighbours of the corresponding probe. The adaptive filter is implemented as a bi-directional LSTM (Schuster & Paliwal, 1997; Graves & Schmidhuber, 2005) and two Fully-Connected layers with shortcuts.

## 3.3 ADA-NETS FOR FACE CLUSTERING

To effectively address the noise edges problem in face clustering, Ada-NETS first takes advantage of the proposed structure space and adaptive neighbour discovery to build a graph with clean and rich edges. Then a GCN model is used to complete the clustering in this graph as shown in Figure 2 (III). With similarity metric in the structure space and adaptive neighbour discovery method above, the discovered neighbours of vertex $\mathbf{v}_i$ is denoted as $\mathcal{N}^s(\mathbf{v}_i, k)$:

$$\mathcal{N}^s(\mathbf{v}_i, k) = \left\{ \mathbf{v}_{i_j} | \mathbf{v}_{i_j} \in \mathcal{N}(\mathbf{v}_i, k), \text{Ind}_j \leq \hat{\mathbf{k}}^{\text{off}} \right\}, \tag{6}$$

where $\text{Ind}_j$ means the index of $\mathbf{v}_{i_j}$ ranked by $\kappa(\mathbf{v}_i, \mathbf{v}_{i_j})$ in the descending order. Based on these neighbour relations, an undirected graph $\mathcal{G}(\mathbf{F}, \mathbf{A})$ is generated by linking an edge between two

vertices if any of which is the discovered neighbour of the other. $\mathbf{F} = [\mathbf{v}_1, \mathbf{v}_2, \cdots, \mathbf{v}_N]^T$ is the vertex feature matrix, and $\mathbf{A}$ is the adjacency matrix:

$$\mathbf{A}_{ij} = \begin{cases} 1, & \mathbf{v}_i \in \mathcal{N}^s(\mathbf{v}_j, k) \text{ or } \mathbf{v}_j \in \mathcal{N}^s(\mathbf{v}_i, k), \\ 0, & \text{otherwise.} \end{cases} \tag{7}$$

With the built graph $\mathcal{G}(\mathbf{F}, \mathbf{A})$, A GCN model is used to learn whether two vertices belong to the same class. One GCN layer is defined as:

$$\mathbf{F}_{l+1} = \sigma(\tilde{\mathbf{D}}^{-1}\tilde{\mathbf{A}}\mathbf{F}_l\mathbf{W}_l), \tag{8}$$

where $\tilde{\mathbf{A}} = \mathbf{A} + \mathbf{I}$, $\mathbf{I} \in \mathbb{R}^{N \times N}$ is an identity matrix, $\tilde{\mathbf{D}}$ is the diagonal degree matrix that $\tilde{\mathbf{D}}_{ii} = \sum_{j=1}^{N} \tilde{\mathbf{A}}_{i,j}$, $\mathbf{F}_l$ and $\mathbf{W}_l$ are respectively the input feature matrix and weight matrix of $l$-th layer, and $\sigma(\cdot)$ is an activation function. Two GCN layers are used in this paper, followed with one FC layer with PReLU (He et al., 2015) activation and one normalization. For a batch of randomly sampled vertices $B_v$, the training loss is defined as a variant version of Hinge loss (Rosasco et al., 2004):

$$\mathcal{L}^{Hinge} = \mathcal{L}^{neg} + \lambda \mathcal{L}^{pos},$$

$$\text{where } \mathcal{L}^{pos} = \frac{1}{\|l_i = l_j\|} \sum_{l_i = l_j} [\beta_1 - y_{\mathbf{v}_i, \mathbf{v}_j}]_+, \quad \mathcal{L}^{neg} = \max_{l_i \neq l_j} [\beta_2 + y_{\mathbf{v}_i, \mathbf{v}_j}]_+, \tag{9}$$

where $y_{\mathbf{v}_i, \mathbf{v}_j}$ is the cosine similarity of the GCN output features $\mathbf{v}'_i$ and $\mathbf{v}'_j$ of the two vertices $\mathbf{v}_i$ and $\mathbf{v}_j$ in the batch $B_v$, $[\cdot]_+ = max(0, \cdot)$, $\|l_i = l_j\|$ is the number of the positive pairs, i.e., the ground-truth label $l_i$ of $\mathbf{v}_i$ and the ground-truth label $l_j$ of $\mathbf{v}_j$ are the same; $\beta_1$ and $\beta_2$ are the margins of the positive and negative losses, and $\lambda$ is the weight balancing the two losses.

During inference, the whole graph of test data is input into GCN to obtain enhanced features $\mathbf{F}' = [\mathbf{v}'_1, \mathbf{v}'_2, \cdots, \mathbf{v}'_N]^T \in \mathbb{R}^{N \times D'}$ for all vertices, where $D'$ is the dimension of each new feature. Pairs of vertices are linked when their similarity scores are larger than a predefined threshold $\theta$. Finally, clustering is done by merging all links transitively with the union-find algorithm, i.e., wait-free parallel algorithms(Anderson & Woll, 1991).

## 4 EXPERIMENTS

### 4.1 EVALUATION METRICS, DATASETS, AND EXPERIMENTAL SETTINGS

The Signal-Noise Rate (SNR) and $Q$-value are used to directly evaluate the quality of graph building, where SNR is the ratio of the number of correct edges and noise edges in the graph. **BCubed F-score** $F_B$ (Bagga & Baldwin, 1998; Amigó et al., 2009) and **Pairwise F-score** $F_P$ (Shi et al., 2018) are used to evaluate the final clustering performance.

Three datasets are used in the experiments: **MS-Celeb-1M** (Guo et al., 2016; Deng et al., 2019) is a large-scale face dataset with about 5.8M face images of 86K identities after data cleaning. For a fair comparison, we follow the same protocol and features as VE-GCN (Yang et al., 2019) to divide the dataset evenly into ten parts by identities, and use part_0 as the training set and part_1 to part_9 as the testing set. In addition to the face data, Ada-NETS has also been evaluated for its potential in clustering other objects. The clothes dataset **DeepFashion** (Liu et al., 2016) is used with the same subset, split settings and features as VE-GCN (Yang et al., 2020), where there are 25,752 images of 3,997 categories for training, and 26,960 images of 3,984 categories for testing. **MSMT17** (Wei et al., 2018) is the current largest ReID dataset widely used (Chen et al., 2021; Jiang et al., 2021). Its images are captured under different weather, light conditions and time periods from 15 cameras, which are challenging for clustering. There are 32,621 images of 1,041 individuals for training and 93,820 images of 3,060 individuals for testing. Features are obtained from a model trained on the training set (He et al., 2020).

The learning rate is initially 0.01 for training the adaptive filter and 0.1 for training the GCN with cosine annealing. $\delta = 1$ for Huber loss, $\beta_1 = 0.9$, $\beta_2 = 1.0$, $\lambda = 1$ for Hingle loss and $\beta = 0.5$ for $Q$-value. The SGD optimizer with momentum 0.9 and weight decay 1e-5 is used. $k$ is set $80, 5, 40$ on **MS-Celeb-1M**, **DeepFashion**, and **MSMT17**. The experiments are conducted with PyTorch (Paszke et al., 2019) and DGL (Wang et al., 2019a).

Table 1: Face clustering performance with different numbers of unlabeled images on MS-Celeb-1M.

| #unlabeled | 584K | | 1.74M | | 2.89M | | 4.05M | | 5.21M | |
|---|---|---|---|---|---|---|---|---|---|---|
| Method | $F_P$ | $F_B$ | $F_P$ | $F_B$ | $F_P$ | $F_B$ | $F_P$ | $F_B$ | $F_P$ | $F_B$ |
| K-Means | 79.21 | 81.23 | 73.04 | 75.20 | 69.83 | 72.34 | 67.90 | 70.57 | 66.47 | 69.42 |
| HAC | 70.63 | 70.46 | 54.40 | 69.53 | 11.08 | 68.62 | 1.40 | 67.69 | 0.37 | 66.96 |
| DBSCAN | 67.93 | 67.17 | 63.41 | 66.53 | 52.50 | 66.26 | 45.24 | 44.87 | 44.94 | 44.74 |
| L-GCN | 78.68 | 84.37 | 75.83 | 81.61 | 74.29 | 80.11 | 73.70 | 79.33 | 72.99 | 78.60 |
| DS-GCN | 85.66 | 85.82 | 82.41 | 83.01 | 80.32 | 81.10 | 78.98 | 79.84 | 77.87 | 78.86 |
| VE-GCN | 87.93 | 86.09 | 84.04 | 82.84 | 82.10 | 81.24 | 80.45 | 80.09 | 79.30 | 79.25 |
| Clusformer | 88.20 | 87.17 | 84.60 | 84.05 | 82.79 | 82.30 | 81.03 | 80.51 | 79.91 | 79.95 |
| DA-Net | 90.60 | - | - | - | - | - | - | - | - | - |
| STAR-FC | 91.97 | 90.21 | 88.28 | 86.26 | 86.17 | 84.13 | 84.70 | 82.63 | 83.46 | 81.47 |
| Ada-NETS | **92.79** | **91.40** | **89.33** | **87.98** | **87.50** | **86.03** | **85.40** | **84.48** | **83.99** | **83.28** |

Table 2: Clustering performance on DeepFashion clothes dataset and MSMT17 person dataset.

| Method | DeepFashion | | MSMT17 | |
|---|---|---|---|---|
| | $F_P$ | $F_B$ | $F_P$ | $F_B$ |
| K-Means | 32.86 | 53.77 | 53.82 | 62.41 |
| HAC | 22.54 | 48.77 | 60.27 | 69.02 |
| DBSCAN | 25.07 | 53.23 | 35.69 | 42.32 |
| Meanshift | 31.61 | 56.73 | 49.22 | 60.06 |
| Spectral | 29.02 | 46.40 | 51.60 | 67.03 |
| L-GCN | 28.85 | 58.91 | 49.19 | 62.06 |
| VE-GCN | 38.47 | 60.06 | 50.27 | 64.56 |
| STAR-FC | 37.07 | 60.60 | 58.80 | 66.92 |
| Ada-NETS | **39.30** | **61.05** | **64.05** | **72.88** |

Table 3: Contribution of the structure space (SS) and adaptive neighbour discovery (AND).

| Method | | Graph Building | Clustering | |
|---|---|---|---|---|
| SS | AND | SNR | $F_P$ | $F_B$ |
| × | × | 1.62 | 77.17 | 77.25 |
| ✓ | × | 2.37 | 81.49 | 84.37 |
| × | ✓ | 13.85 | 89.29 | 88.98 |
| ✓ | ✓ | **21.67** | **92.79** | **91.40** |

## 4.2 METHOD COMPARISON

Face clustering performance is evaluated on the **MS-Celeb-1M** dataset with different numbers of unlabeled images. Comparison approaches include classic clustering methods K-Means (Lloyd, 1982), HAC (Sibson, 1973), DBSCAN (Ester et al., 1996), and graph-based methods L-GCN (Wang et al., 2019b), DS-GCN (Yang et al., 2019), VE-GCN (Yang et al., 2020), DA-Net (Guo et al., 2020), Clusformer (Nguyen et al., 2021) and STAR-FC (Shen et al., 2021). In this section, to further enhance the clustering performance of GCNs, some noise is added to the training graph. Results in Table 1 show that the proposed Ada-NETS reaches the best performance in all tests ($\theta = 0.96$), outperforming STAR-FC by 1.19% to reach 91.40% BCubed F-score on 584K unlabeled data. With the increase in the number of unlabeled images, the proposed Ada-NETS keeps superior performance, revealing the significance of graph building in large-scale clustering.

To further evaluate the generalization ability of our approach in non-face clustering tasks, comparisons are also conducted on **DeepFashion** and **MSMT17**. As shown in Table 2, Ada-NETS achieves the best performance in the clothes and person clustering tasks.

## 4.3 ABLATION STUDY

**Study on the structure space and adaptive neighbour discovery** Table 3 evaluates the contributions of the structure space and adaptive neighbour discovery. The SNR of the built graph and the BCubed and Pairwise F-scores are compared. It is observed that the structure space and adaptive neighbour discovery both contribute to the performance improvement, between which the adaptive neighbour discovery contributes more. With both components, the graph's SNR is largely enhanced

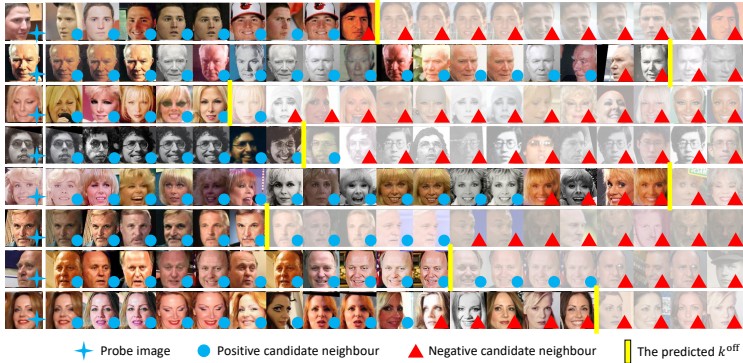

Figure 4: Random examples of top 20 images ranked by the similarity with probe images.

Table 4: Graph building and clustering performance of the quality criterion with different $\beta$.

| # | $\beta$ | Graph Building | | Clustering | |
|---|---|---|---|---|---|
| | | $Q^{\text{before}}$ | $Q^{\text{after}}$ | $F_P$ | $F_B$ |
| | 0.5 | 69.16 | 82.98 | **91.64** | **91.62** |
| Ori. | 1.0 | 67.94 | 72.72 | 88.13 | 87.45 |
| | 2.0 | 66.76 | 67.53 | 84.78 | 84.43 |
| | 0.5 | 69.16 | 88.96 | **95.51** | **94.93** |
| Str. | 1.0 | 67.94 | 77.18 | 94.54 | 93.97 |
| | 2.0 | 66.76 | 69.38 | 94.16 | 93.14 |

Table 5: Different estimation methods of the adaptive filter.

| Method | Target | $F_P$ | $F_B$ |
|---|---|---|---|
| GAT | - | 81.38 | 81.46 |
| $\mathbf{E}^{\mathbf{Q}}_{\text{seq}}$ | $Q$ | 86.10 | 87.61 |
| $\mathbf{E}^{\mathbf{Q}}_{\text{param}}$ | $Q$ | 59.56 | 68.05 |
| $\mathbf{E}^{\mathbf{k}}_{\text{cls}}$ | $\mathbf{k}^{\text{off}}$ | 92.55 | **91.63** |
| $\mathbf{E}^{\mathbf{k}}_{\text{reg}}$ | $\mathbf{k}^{\text{off}}$ | **92.79** | 91.40 |

by 13.38 times, and the clustering performance is also largely improved. Each line in Figure 4 shows the discovery results with the first image as the probe, ranked by the similarity in the structure space. The images with a blue circle are of the same identity as the probe. They all successfully obtain greater similarity in the structure space than those with a red triangle that represents different identities. With the help of the adaptive filter, images after the yellow vertical line are filtered out, remaining clean and rich neighbours. Without the adaptive filter, the images with a red triangle will be connected with its probe, leading to pollution of the probe.

**Study on the quality criterion** According to Equation 3, the criterion contains a hyper-parameter $\beta$. Smaller $\beta$ emphasizes more on precision, and bigger $\beta$ emphasizes more on recall. We choose three mostly-used values $\beta \in \{0.5, 1.0, 2.0\}$ to study how it affects the neighbour discovery and clustering. Table 4 shows the performance of Ada-NETS under different $\beta$ in the original (denoted as Ori.) and structure space (denoted as Str.). $Q^{\text{before}}$ and $Q^{\text{after}}$ are the $Q$-value of candidate neighbours of size $k$ and $k^{\text{off}}$. $F_P$ and $F_B$ are the corresponding clustering performance under $k^{\text{off}}$. It is observed that $Q^{\text{after}}$ is obviously improved compared with $Q^{\text{before}}$ in all circumstance. For the same $\beta$, the improvements are more obvious in the structure space than in the original space. As analysed above, a higher $Q$-value indicates a better candidate neighbours quality, e.g., more noise edges will be eliminated (clean) or more correct edges will be kept (rich) in the graph. Therefore, the clustering performance in the structure space is also higher than in the original space as is expected. In addition, $\beta = 0.5$ achieves the best clustering performance in both spaces while much less sensitive in the structure space to reach 95.51% Pairwise F-score and 94.93% BCubed F-score.

**Study on the adaptive filter** Adaptive filter is proposed to select neighbours from candidate neighbours. Compared with the estimation method to regress $\mathbf{k}^{\text{off}}$ directly in the adaptive filter, noted as $\mathbf{E}^{\mathbf{k}}_{\text{reg}}$, some other estimation methods are also studied: $\mathbf{E}^{\mathbf{k}}_{\text{cls}}$ formulates the $\mathbf{k}^{\text{off}}$ estimation as a $k$-class classification task; $\mathbf{E}^{\mathbf{Q}}_{\text{seq}}$ regresses $Q$-value for all $j$ directly; $\mathbf{E}^{\mathbf{Q}}_{\text{param}}$ fits $Q$-value with respect to $j$ with a quadratic curve and estimates the parameters of this curve. Results in Table 5 show that $\mathbf{E}^{\mathbf{k}}_{\text{reg}}$ and $\mathbf{E}^{\mathbf{k}}_{\text{cls}}$ that estimate $k^{\text{off}}$ obtain obviously higher performance than $\mathbf{E}^{\mathbf{Q}}_{\text{seq}}$ and $\mathbf{E}^{\mathbf{Q}}_{\text{param}}$ that estimate the $Q$-value. Compared with $\mathbf{E}^{\mathbf{k}}_{\text{cls}}$, $\mathbf{E}^{\mathbf{k}}_{\text{reg}}$ achieves close results but needs less parameters

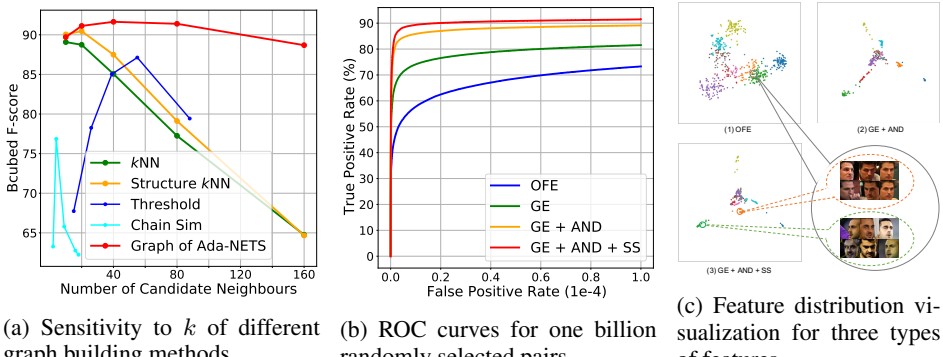

(a) Sensitivity to $k$ of different graph building methods.

(b) ROC curves for one billion randomly selected pairs.

(c) Feature distribution visualization for three types of features.

Figure 5: (a) The sensitivity of clustering to $k$. Ada-NETS maintains a stable and outstanding performance. (b) The ROC curves on MS-Celeb-1M part_1. All embedded features have better ROC performances than the original feature embedding, and the graph embedding output by GCN with the help of AND and SS has the best performance. (c) Feature distribution visualization for three types of features: original feature embedding (c.1), graph embedding of GCN with the adaptive neighbour discovery strategy in the original space (c.2) and structure space (c.3).

to learn. GAT is also compared to eliminate noise edges by the attention mechanism, but does not obtain competitive results because of the complex feature pattern (Yang et al., 2020).

**Study on the sensitivity to $k$** Figure 5 (a) shows the clustering performance with the variance of $k$. "$k$NN" represents directly selecting the nearest $k$ neighbours to build the graph as in existing methods (Yang et al., 2020; Shen et al., 2021), and "Structure $k$NN" represents selecting $k$NN in the structure space with $\mathcal{N}(\mathbf{v}, 256)$. Despite the help in the structure space, the two $k$NN methods are all sensitive to $k$ since more noise edges are included when $k$ increases. The "Threshold" and "Chain Sim" method are also also sensitive to the number of candidate neighbours and have not achieved good results. The details of each method can be found in Table 9 in appendix. However, the proposed Ada-NETS can relatively steadily obtain good performance, showing that our method can effectively provide clean and rich neighbours to build the graph for GCNs.

**Study on the graph embedding** In Ada-NETS, GCN is used to produce more compactly distributed features, thus more suitable for clustering. ROC curves of one billion randomly selected pairs are shown in Figure 5 (b). It is observed that the original feature embedding (OFE) obtains the worst ROC performance, and the graph embedding (GE) output by the GCN is enhanced by a large margin. With the help of the adaptive neighbour discovery (GE + AND), the output feature is more discriminating. When the discovery is applied in the structure space (GE + AND + SS), GCN can output the most discriminating features. Distributions of embeddings after dimensionality reduction via PCA (Pearson, 1901) of ten randomly selected identities are shown in Figure 5 (c). It is observed that the better ROC performance a kind of feature embedding has in Figure 5 (b), the more compact its embeddings are for a certain identity. With better features embeddings, the clustering can be started again. In Table 10 of appendix, the clustering results on MS-Celeb-1M are compared with the ones using original feature embeddings. It is observed that the graph embeddings further enhance Ada-NETS. The details are described in Section A.8 of appendix.

## 5 CONCLUSION

This paper presents a novel Ada-NETS algorithm to deal with the noise edges problem when building the graph in GCN-based face clustering. In Ada-NETS, the features are first transformed to the structure space to enhance the accuracy of the similarity metrics. Then an adaptive neighbour discovery method is used to find neighbours for all samples adaptively with the guidance of a heuristic quality criterion. Based on the discovered neighbour relations, a graph with clean and rich edges is built as the input of GCNs to obtain state-of-the-art on the face, clothes, and person clustering tasks.

**Acknowledgement** This work was supported by Alibaba Group through Alibaba Innovative Research Program.

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

## A  APPENDIX

In this appendix, we present the supplementary information on 1) the calculations of the precision and recall; 2) the calculation of the Jaccard similarity; 3) the detailed description of different designs of the adaptive filter; 4) the baseline results of selecting neighbours by various thresholds based on the cosine similarity of deep features; 5) the baseline results of directly clustering on the built graph without GCNs; 6) the time-consuming analysis of Ada-NETS.

### A.1  PRECISION AND RECALL

In the paper, precision $Pr^j$ and recall $Rc^j$ are used to obtain the quality criterion $Q$ in Equation 3. The calculations of $Pr^j$ and $Rc^j$ are illustrated by example in Figure 6. Given a dataset that has 15 samples of three classes, assume that one probe sample belonging to class 1 is selected from the dataset. Samples are first ranked by their similarities to the probe. For candidate neighbours of size $j$ ($j = 6$), $Pr^j$ and $Rc^j$ are $\frac{4}{6}$ and $\frac{4}{8}$ respectively, as calculated in the Figure 6.

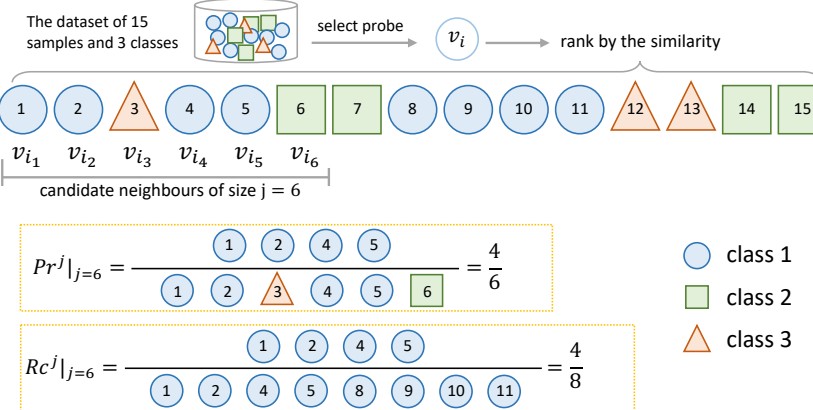

Figure 6: The calculation of precision $Pr^j$ and recall $Rc^j$.

### A.2  THE JACCARD SIMILARITY

In the paper, feature similarities are calculated in the structure space as in Equation 2. This similarity includes the Jaccard similarity which is calculated based on the $k$-reciprocal nearest neighbours (Zhong et al., 2017), defined as

$$s^{\text{Jac}}\left(\mathbf{v}_i, \mathbf{v}_{i_j}\right) = \frac{|\mathcal{R}^*(\mathbf{v}_i, k) \cap \mathcal{R}^*(\mathbf{v}_{i_j}, k)|}{|\mathcal{R}^*(\mathbf{v}_i, k) \cup \mathcal{R}^*(\mathbf{v}_{i_j}, k)|}, \tag{10}$$

where $\mathcal{R}^*(\mathbf{v}, k)$ encodes the structure information of feature $\mathbf{v}$ with a variant version of $k$-reciprocal nearest neighbours:

$$\mathcal{R}^*(\mathbf{v}, k) \leftarrow \mathcal{R}(\mathbf{v}, k) \cup \mathcal{R}(\mathbf{r}, \frac{1}{2}k),$$

$$\text{s.t. } |\mathcal{R}(\mathbf{v}, k) \cap \mathcal{R}(\mathbf{r}, \frac{1}{2}k)| \geq \frac{2}{3}|\mathcal{R}(\mathbf{r}, \frac{1}{2}k)|, \ \forall \mathbf{r} \in \mathcal{R}(\mathbf{v}, k), \tag{11}$$

where

$$\mathcal{R}(\mathbf{v}, k) = \{\mathbf{r} \mid \mathbf{v} \in \mathcal{N}(\mathbf{r}, k), \ \forall \mathbf{r} \in \mathcal{N}(\mathbf{v}, k)\}. \tag{12}$$

With the definition above, data distribution can be perceived in the structure space to obtain more accurate feature similarities.

### A.3 Designs of adaptive filter

In the paper, four kinds of different designs for the adaptive filter are proposed and their performances are compared in Table 5. Two methods estimate $Q$ values, and two estimate $\mathbf{k}^{\text{off}}$ directly. Table 6 shows detailed descriptions of the four designs.

Table 6: Descriptions of four kinds of different designs of the adaptive filter.

| Symbol | Method Description |
|---|---|
| $\mathbf{E}^{\mathbf{Q}}_{\mathbf{seq}}$ | Regress $Q$-values in Equation 3 in the paper for each $j \in \{1, 2, \cdots, k\}$. During inference, $\hat{\mathbf{k}}^{\text{off}}$ is determined by Equation 4 using predicted $Q$-values. |
| $\mathbf{E}^{\mathbf{Q}}_{\mathbf{param}}$ | Use a quadratic curve to fit the trends of $Q(j)$ with respect to $j$, which is then used to predict $Q$ values to get $\hat{\mathbf{k}}^{\text{off}}$ as defined in Equation 4 during inference. The parameters of the quadratic curve is learned as targets in this method. |
| $\mathbf{E}^{\mathbf{k}}_{\mathbf{cls}}$ | Formulate the $\mathbf{k}^{\text{off}}$ estimation as a multi-class classification task. One class for each possible $j$ value in Equation 4 and use softmax loss in training. |
| $\mathbf{E}^{\mathbf{k}}_{\mathbf{reg}}$ | Regress directly the ground-truth $\mathbf{k}^{\text{off}}$ values as defined in Equation 4. |

### A.4 Clustering with fixed thresholds

In addition to directly selecting the nearest $k$ neighbours to build the graph as in Figure 5 (a), selecting nearest neighbours by a fixed similarity threshold is also evaluated. In Table 7, it is observed that the best performance is achieved when the similarity threshold is $0.60$. The highest Pairwise and BCubed F-score are 87.59 and 87.13, which are much lower than 92.79 and 91.40 of Ada-NETS. This shows that the method of selecting neighbours by a fixed threshold for all samples is not as good as the proposed method in Ada-NETS, which is adaptive for each sample.

Table 7: Bcubed F-score of using various thresholds to select neighbours to build graph.

| Threshold | 0.75 | 0.70 | 0.65 | 0.60 | 0.55 |
|---|---|---|---|---|---|
| $F_P$ | 68.48 | 79.64 | 86.58 | **87.59** | 77.88 |
| $F_B$ | 67.74 | 78.26 | 85.04 | **87.13** | 79.43 |

### A.5 Clustering without GCNs

When a graph with clean and rich edges is built after neighbour discovery, a naive graph-cut can be conducted on the graph as the clustering baseline. The graph-cut method directly breaks the edges that have similarities lower than a predefined similarity threshold (which is tuned for the best), and no GCNs are used.

Table 8: Comparison of clustering performance of graph-cut (in the original space and structure space) and Ada-NETS on MS-Celeb-1M part_1.

| Method | $F_P$ | $F_B$ |
|---|---|---|
| Graph-cut (Original) | 72.95 | 76.22 |
| Graph-cut (Structure) | 82.36 | 85.09 |
| Ada-NETS | **92.79** | **91.40** |

In Table 8, the first two lines show the clustering performance of graph-cut in the original space and structure space. Compared with Ada-NETS, the first two methods have very poor performances. This shows that the GCN module is important for clustering. In addition, it can be seen that the clustering performance using graph-cut in the structure space is better than in the original space, proving

that vertices can obtain robust features by encoding more texture information after perceiving the data distribution in the structure space.

## A.6 CLUSTERING TIME-CONSUMING ANALYSIS

Empirically, Ada-NETS takes about 18.9 minutes to cluster part_1 test set (about 584k samples) on 54 E5-2682 v4 CPUs and 8 NVIDIA P100 cards. It is much faster than L-GCN (Wang et al., 2019b) and LTC (Yang et al., 2019) which take 86.8 minutes and 62.2 minutes, but slower than STAR-FC (Shen et al., 2021), DA-NET (Guo et al., 2020) and VE-GCN (Yang et al., 2020) which take about 5 to 11 minutes. Many parts of Ada-NETS can be easily parallelized, but it is out of the scope of this work. Although Ada-NETS is not the fastest, the time consumption is still within a reasonable range.

## A.7 COMPARISON WITH OTHER GRAPH BUILDING METHODS

In Figure 5 (a), the Bcubed F-score of Ada-NETS is compared with four other graph building methods under different $k$ value to show its superiority and robustness. Methods for comparison include: $k$NN, Structure $k$NN, Threshold, and Chain Sim. Table 9 shows detailed description of these graph building methods.

Table 9: Descriptions of different graph building methods.

|  | Parameter Setting | Method Description |
|---|---|---|
| $k$NN | $k$ is set at 10, 20, 40, 80 and 160. | Each vertex is connected with its $k$ nearest neighbours as in Yang et al. (2019; 2020). |
| Structure $k$NN | $k$ is set at 10, 20, 40, 80 and 160. | Each vertex is connected with its $k$ nearest neighbours in the structure space proposed in the paper. |
| Threshold | Vary the threshold from 0.55 to 0.75 by step of 0.05. $k$ is calculated as the mean of the neighbours for each vertex at the given threshold. | Choose neighbours whose cosine similarities to the probe vertex are smaller than a predefined threshold as in Guo et al. (2020). |
| Chain Sim | Vary the similarity threshold from 0.4 to 0.8 by step of 0.1. $k$ is calculated as the mean of the number of neighbours for each vertex at the given threshold. | First calculate each vertex's density, and then construct a chain by incrementally adding the vertices which have similarity scores higher than a predefined threshold and also densities larger than the last one as in Guo et al. (2020). |
| Graph of Ada-NETS | The candidate neighbour size is set at 10, 20, 40, 80 and 160. | Build the graph with the method proposed in Ada-NETS. |

## A.8 ADA-NETS WITH BETTER FEATURES EMBEDDINGS

Table 10: Pairwise F-score of original feature embedding and graph embedding on MS-Celeb-1M.

| Input Feature | 584K | 1.7M | 2.9M | 4.1M | 5.2M |
|---|---|---|---|---|---|
| OFE | 92.79 | 89.33 | 87.50 | 85.40 | 83.99 |
| GE+AND+SS | **93.74** | **91.19** | **89.42** | **87.91** | **85.65** |

These discriminating graph embeddings of "GE + AND + SS" are used as the input of Ada-NETS to get the final clustering results for enhancement ($\theta = 0.99$). In Table 10, the clustering results on MS-Celeb-1M are compared with the ones using original feature embeddings. It is observed that the

graph embeddings further enhance Ada-NETS from state-of-the-art to a remarkable 93.74% on $F_P$, achieving nearly 1% improvement on 584K unlabeled data again.

