# OpenReview forum: "Ada-NETS: Face Clustering via Adaptive Neighbour Discovery in the Structure Space"
_ICLR.cc/2022/Conference — ICLR 2022 Poster_

### Official Review · Reviewer_FhxA · 2021-10-21

**Correctness:** 3
**Technical Novelty And Significance:** 3
**Empirical Novelty And Significance:** 2
**Recommendation:** 6
**Confidence:** 4

**Main Review:**

The paper provides a new perspective to boost the performance of GCN-based face clustering methods.
In summary, the novelty is good and experimental result seems promising.
But the experimental evaluation needs more explaination and  extension.

**Summary Of The Paper:**

This paper proposes a novel algorithm named Ada-NETS to alleviate the noise edges problem when building a graph on face images, thus improve GCNs to boost face clustering performance.
Strengths:
1) This is the first paper to address the noise edges problem when building a graph for GCNs on face images.
2) The proposed Ada-NETS achieves state-of-the-art performances on multiple public datasets. And the experimentation is vert sufficient.
weaknesses:
The author chooses F-beta-score as the candidate neighbors quality criterion, which seems to be a little bit heuristic. It is expected to explain the motivation of this design and demonstrate its superiority over other quality criterions.

**Summary Of The Review:**

The experimental evaluation is unsatisfatory.

---

> ### Author Response · Authors · 2021-11-21
> **Response to Reviewer FhxA**
>
>
> We sincerely thank the reviewer for appreciating the novelty and sufficiency of the experiments in our paper. About the proposed concerns, we will respond to them one by one as follows.
>
> $\textbf{Q1. [The motivation and superiority of the quality criterion.]}$
>
> The motivation of the proposed criterion comes from the following two very intuitive principles:
>
> (1) Good neighbours should be clean, i.e., most neighbours should have the same class label with the probe vertex, so that noise edges will not be included in large numbers when building a graph.
>
> (2) The neighbours should also be rich, so that the message can fully pass in the graph.
>
> To satisfy the two principles, the criterion $Q$ is designed.
> The calculation process of precision and recall in Equation (3) is illustrated in Figure 6.
>
> Regarding the superiority of quality criterion, we studied it by $\beta$.
> When $\beta$ approaches 0, the $Q$ is equivalent to precision.
> When $\beta$ approaches the positive infinity, the $Q$ is equivalent to recall.
> Different $\beta$ represents different criteria.
> We carefully compare three mostly-used values $\beta \in \\{0.5, 1.0, 2.0\\}$ (three kinds of criteria) in Table 4 to make sure the superiority of the criterion used.
>
>
>
> $\textbf{Q2. [The experimental evaluation needs more explanation and extension.]}$
>
> Following the comments of the reviewer, we make extensions in the experiments and add more explanations as follows.
>
> (1) We also validated the performance of Ada-NETS under features of various qualities.
> We use ResNet-18, ResNet-50, ResNet-101 to extract features of different qualities.
> It shows that Ada-NETS works the best on all kinds of features, proving it can be less sensitive to the feature quality.
> Please refer to our comments to $Q2$ of the Reviewer Q8yu for more details.
>
> (2) We added two more graph construction methods~(Threshold, Chain Sim) to compare with graphs of Ada-NETS. It is already updated in Figure 5(a) in paper and the detailed process are in the newly added Table 10.
> Among the 5 graph construction methods in Figure 5(a), the proposed graph building in Ada-NETS achieves the best performance and shows its robustness to different $k$ settings (under different noise conditions).
> Please also refer to our comments to $Q3$ of the Reviewer DwFA.

---

### Official Review · Reviewer_MCAC · 2021-10-31

**Correctness:** 4
**Technical Novelty And Significance:** 2
**Empirical Novelty And Significance:** 3
**Recommendation:** 3
**Confidence:** 4

**Main Review:**

There a two main issues with the paper: The first one is the limited amount of technical contribution for a high quality venue like ICLR. The authors put a lot of existing techniques together to build their algorithm but overall the proposed combination as a whole does not constitute a significant and widely applicable contribution to the field that might be of interest to the ICLR community. Secondly, the application domain is primarily limited to faces which is very specific application domain. On the positive side the experiments show promising performance and ablation studies present interesting results. I believe that CVPR is a much better fit for this paper where the contribution and application domain will be much more appreciated by the corresponding community.

**After Rebuttal:** I would like to thank the reviewers for their response. Unfortunately, I still believe that the paper's focus (face clustering) is too narrow for ICLR and that the novelty is limited as a number of existing techniques are put together to solve a very domain-specific problem.

**Summary Of The Paper:**

This paper proposes a new way of constructing a graph for Graph Convolutional Networks (GCNs) for face clustering that is claimed to alleviate the problem of having a significant amount of noisy edges in the graph as in previous methods for face clustering. Their method consists of two innovations: firstly, facial features are transformed to a structured space which improevs the accuracy of facial similarity metrics. Secondly, an method which finds the neighbours of each face is proposed based on a heuristic quality criterion. Once the graph is built the authors apply existing algorithms for GCNs.

**Summary Of The Review:**

A decent paper with good amount of novelty for a computer vision conference where faces are particularly studied but of limited novelty and interest for the ICLR community.

---

> ### Author Response · Authors · 2021-11-21
> **Response to Reviewer MCAC**
>
> We sincerely thank the reviewer for "A decent paper with good amount of novelty for a computer vision conference". About the proposed concerns in three aspects, we will respond to them one by one as follows.
>
> $\textbf{Q1. [About the technical contribution for the ICLR community]}$
>
> (1) To the best of our knowledge, this is the first paper to address the noise edges problem when building a graph for GCNs on face images.
> Simultaneously, this paper demonstrates its causes, great impact, weaknesses in existing solutions and the challenges to solve it.
> We present our solutions corresponding to these challenges, which are logical and effective.
>
> (2) To solve the noise edges problem above, several technical innovations are put forward.
> Four types of estimation methods for selecting neighbours are proposed and compared in Table 5, whose detailed processes are in Table 7. These methods cover the prediction of $Q$ and $\textbf{k}^{\text{off}}$ from the perspective of regression and classification.
> The proposed criterion of $Q$ evaluates the quality of candidate neighbours for each probe vertex. The extreme point of the $Q(j)$ curve is used to select neighbours.
> To our knowledge, this is the first paper to construct a graph from this perspective.
>
> (3) Several contributions are also made for the best performance of Ada-NETS.
> A variant version of Hinge loss in Equation 9 is used to control the distance between features of the same or different classes, aiming for a reasonable threshold for clustering.
> Huber loss in Equation 5 is adopted to handle the problem of imbalance distribution in the regression.
> The $\beta$ in $Q$ is also studied in original and structure space in Table 4.
> it is impossible to reach such a remarkable performance by simply putting the proposed modules together.
>
>
> $\textbf{Q2. [Face clustering is a specific application domain and ICLR may not be interested in it]}$
>
> (1) Face clustering is an important task in computer vision.
> It has attracted increasing research interests in CV due to its importance and wide applications as shown in the 2nd paragraph in the Introduction.
> It can help to manage face photos with fewer human involvements, show the internal structure of the datasets, prepare for the downstream AI techniques, and have potential application value in other fields.
> Other data analysis methods can also be inspired by our method.
>
> (2) As the ICLR conference officially announced at the bottom of this page  (https://iclr.cc/Conferences/2022),  "ICLR is globally renowned for presenting and publishing cutting-edge research on all aspects of deep learning, ... as well as $\textbf{important application areas such as machine vision}$", and "We consider a broad range of subject areas including ... $\textbf{applications in vision}$"(https://iclr.cc/Conferences/2022/CallForPapers).
> As some researchers count, computer vision is already ranked 13th on the list of the ICLR 2022 submission top 50 keywords on Github.
> Therefore, we believe that the important application in computer vision, face clustering, is of enough interest to the ICLR community.
>
> (3) The ICLR community has always welcomed application papers in various fields, such as:
> the paper for biological sequence design in ICLR2020 (Angermueller et al., 2020),
> the paper for molecular conformation generation in ICLR2021 (Xu et al., 2021) ,
> the paper for the toolkit for medical time series in ICLR2021 (Jarrett et al., 2021),
> the paper for the data-driven traffic forecasting ICLR2018 (Li et al., 2018), and so on.
> So we believe that our face clustering paper which is an important application contribution in computer vision should not be excluded.
>
> (4) As shown in Figure 5 (b)(c), the proposed Ada-NETS essentially clusters faces by letting the GCN learn better representation, which is very much in line with the theme of the ICLR meeting.
>
> (5) To further alleviate the reviewer’s concern about the application domain, we also evaluate our approach in non-face clustering tasks, DeepFashion (clothes) and MSMT17 (persons). The results in Table 2 prove the generalization ability of Ada-NETS.
>
> reference:
>
> [1] Angermueller, Christof, et al. "Model-based reinforcement learning for biological sequence design." ICLR (2019).
>
> [2] Roblem et al. "Learning neural generative dynamics for molecular conformation generation." ICLR (2021).
>
> [3] Jarrett, Daniel et al. "Clairvoyance: A Pipeline Toolkit for Medical Time Series." ICLR (2021).
>
> [4] Yaguang Li, Rose Yu et al. "Diffusion Convolutional Recurrent Neural Network: Data-Driven Traffic Forecasting" ICLR (2018).

---

### Official Review · Reviewer_Q8yu · 2021-11-03

**Correctness:** 4
**Technical Novelty And Significance:** 4
**Empirical Novelty And Significance:** 4
**Recommendation:** 8
**Confidence:** 4

**Main Review:**

**StrongPoints**

(1) In the reviewer's perspective, the proposed design is well organized to enhance the clustering performance.

(2) The datasets, MS1M, deep-fashion, and MSMT17, used in the experiments are large and realistic.

(3) The experimental results in Table 1 show the method's superiority compared to current state-of-the-art methods.

(4) Table three to six provides analyses on each proposion of the paper, which is well organized.


**WeakPoints**

First, note that the reviewer is positive about this work, and this section is much like questions rather than criticism.

(1) It seems that the results in Table 1 would be governed by the expression power of the face feature v. The reviewer might miss and thank the author in advance for the clarification of the feature extraction process for the experiment. Similar to the other datasets in the experiment.

(2) Also, since it is possible to access face features with various representation powers, the reviewer wants to see the tendency between face recognition performance and face clustering. For example, if the user uses powerful recognition features or vice versa, how does the effect of the proposed contribution change? It would be nice to refer to the webpage here: https://github.com/deepinsight/insightface/tree/master/model_zoo.

(3) Does the author have a plan to open the source code of the paper? It would be helpful for all the researchers in this field.


**Overall**

The reviewer is positive about the paper. Unless some critical issues during the reviewing process occur (such as missing very important s-o-t-a methods), and if detailed comments of the authors are provided, the reviewer thinks the paper is worth publishing in this stage.


**Post Rebuttal**
Thank you for the rebuttal. The reviewer is satisfied with the feature extraction process and experiments on features with various recognition performances. It seems that the numbers in the Tables show the feature robustness of the proposed Ada-NETS. Overall the reviewer is positive for the usefulness of the method and will keep the current rating (accept. Good paper).


**Summary Of The Paper:**

This paper proposes a new face clustering algorithm that significantly improves the past state-of-the-art GCN, which generates many noise edges that are harmful to clustering performance. The paper focuses on the noise edges in face clustering fields and proposes Adaptive Neighbour discovEry in the strucTure Space (Ada-NETS). In Ada-NETS, a 'structure space' is presented to obtain robust face features by encoding more texture information reflecting the data distribution. Then, a candidate neighbor quality criterion is presented to build less-noisy edges, along with a learnable adaptive filter to learn this criterion. Finally, the algorithm's effectiveness is validated by measuring the performance on various data domains: face, person, and clothes, with a large margin.

Technically,

(1) The paper strengthens the credibility of the edge between vertices by applying the Jaccard similarity of the two vertices proposed by Zhong et al., 2017, with widely used cosine similarity.

(2) To design adaptive neighbor discovery, they first define a quality criterion reflecting the two quality factors: cleanness and information abundance (rich) of the graph.

(3)  Using top-k quality criterion vertices as gt, adaptive filter defined by bidirectional lstm and fully connected layers is trained to predict a future quality estimate to remove or add the neighbors.

**Summary Of The Review:**

See the section above.

---

> ### Author Response · Authors · 2021-11-21
> **Response to Reviewer Q8yu**
>
>
>
> We sincerely thank the reviewer for the appreciation of our paper, including the organization and "the method's superiority". About the proposed concerns, we will respond to them one by one as follows.
>
> $\textbf{Q1. [The feature extraction process for the experiments]}$
>
> For a fair comparison, we follow the same feature extraction protocol as in DS-GCN (Yang et al., 2019) and VE-GCN (Yang et al., 2020) for the MS-Celeb-1M and DeepFashion datasets: Given a face dataset, we extract the features for each face image with a trained CNN, forming a set of features $\mathcal{D} = \\{f_i\\}^{N}_{i=1}$, where $f_i$ is a $d$-dimensional vector. For example, for the MS-Celeb-1M dataset, the ResNet-50 CNN model is trained on part0 with softmax cross-entropy loss.
> The feature files of MS-Celeb-1M and DeepFashion are already publicly available on GitHub.
> For the features of the MSMT17 dataset, we trained a ResNet-50 CNN model on the training set and extract all features by it. FastReid (He et al., 2020)  is used as the training framework.
>
>
> $\textbf{Q2. [The tendency between face recognition performance and face clustering.]}$
>
> (1) Thanks for the advice from the reviewer, and we have checked the model zoo site of insightface. These SOTA models are all trained on public celebrity sets, such as Glint360K, WebFace600K, and MS-Celeb-1M. However, the identities of these sets overlap with our clustering test set on a large scale. Evaluating clustering performance using features extracted from these models will lead to unfair comparisons. As an alternative, we designed the following experiment to answer the reviewer’s questions as in (2).
>
> (2) As the reviewer wants to see the tendency between face recognition performance and face clustering, we trained three models (ResNet-18, ResNet-50, and ResNet-101) on MS-celeb part0 by the HFsoftmax pipeline (Zhang, at al, 2018) with default settings, and use these models to extract features for experiments.
> The three models are evaluated on LFW, CFP-FP, and AgeDB-30.
> We further evaluated the features' retrieval performance with the precision at a given number of recalled neighbours (Prec@1, Prec@5, Prec@50, Prec@100) on part1.
> The evaluations of the three models are as follows in Table R1.
>
> Table R1. The evaluation of ResNet18, ResNet50 and ResNet101
>
> |             | LFW    | CALFW  | CPLFW  | Prec@1  | Prec@5  | Prec@50  | Prec@100 |
> |-------------|:------:|:------:|:------:|:------:|:------:|:------:|:-------:|
> | ResNet\-18  | 98\.08 | 90\.92 | 83\.93 | 95\.74 | 91\.52 | 71\.78 | 54\.23  |
> | ResNet\-50  | 98\.69 | 91\.62 | 84\.88 | 97\.05 | 94\.25 | 77\.58 | 59\.79  |
> | ResNet\-101 | 98\.72 | 92\.52 | 85\.47 | 97\.40 | 94\.97 | 79\.52 | 61\.82  |
>
> It can be observed from Table R1 that the ResNet-101 works the best. The ResNet-18 works the worst and the performance of ResNet-50 is between the two.
>
> We choose three clustering methods: Ada-NETS, K-means and clustering by one tuned threshold on original features (denoted Thres-Cut for short).
> K-means works the best in the classic clustering methods in our experiments.
> We conduct experiments on these three-level features, and the results are as follows.
>
> Table R2. The clustering performance of Ada-NETS.
>
> |             | BCubed F\-Score | Pairwise F\-score |
> |-------------|:---------------:|:-----------------:|
> | ResNet\-18  | 84\.69          | 84\.48            |
> | ResNet\-50  | 89\.50          | 90\.77            |
> | ResNet\-101 | 90\.57          | 91\.84            |
>
> Table R3. The clustering performance of Thres-Cut.
>
> |             | BCubed F\-Score | Pairwise F\-score |
> |-------------|:---------------:|:-----------------:|
> | ResNet\-18  | 61\.91          | 57\.38            |
> | ResNet\-50  | 68\.65          | 63\.40            |
> | ResNet\-101 | 73\.23          | 70\.92            |
>
> Table R4. The clustering performance of K-means.
>
> |             | BCubed F\-Score | Pairwise F\-score |
> |-------------|:---------------:|:-----------------:|
> | ResNet\-18  | 73\.91          | 72\.74            |
> | ResNet\-50  | 79\.26          | 77\.82            |
> | ResNet\-101 | 80\.91          | 79\.36            |
>
> It can be observed that:
>
> $\bullet$ The clustering performance of Ada-NETS increases with the improvement of feature quality in Table R2.
>
> $\bullet$ The clustering performance gap between Ada-NETS and Thres-Cut decreases with the improvement of feature quality, comparing Table R2 and Table R3.
>
> $\bullet$ Among the three methods, Ada-NETS works the best on all kinds of feature qualities, showing it can be less sensitive to the feature quality.
>
>
> $\textbf{Q3. [About the source code.]}$
>
> We are glad to contribute to the ICLR community and will release the code and datasets after the paper is accepted. The experiments above will be considered and added to the appendix in the camera-ready version after accepted.

---

### Official Review · Reviewer_DwFA · 2021-11-04

**Correctness:** 2
**Technical Novelty And Significance:** 3
**Empirical Novelty And Significance:** 3
**Recommendation:** 6
**Confidence:** 5

**Main Review:**

strengths:
1. The paper is easy to follow;
2. Experimental results are good.
weaknesses:
1. The novelty of the proposed method is limited. Both of the Eq. (2), Eq. (3) and the GCN structure are previously proposed, this paper just combines them to obtain better results;
2. Since the task of this paper is clustering, which is an unsupervised task. However, but in Eq. (3), sample labels are used to calculate quality criterion Q, which is not clear.
3. The adaptive neighbour discovery and GCN are two seperated processes in current propose method. Therefore, any other graph optimization method can be used before the GCN process. The authors should conduct experiments to validate the efficacy of the proposed adaptive graph construction.

**Summary Of The Paper:**

In order to address the problem that face clustering is often degraded by noise edges in similarity graph, this paper proposes Ada-NETS by constructing clean graphs for GCNs. In Ada-NETS, each face is transformed to a new structure space, obtaining robust features by considering face features of the neighbour images. Then, an adaptive neighbour discovery strategy is proposed to determine a proper number of edges connecting to each face image. Experiments on multiple public clustering datasets show that Ada-NETS significantly outperforms current state-of-the-art methods.

**Summary Of The Review:**

Novelty is limited and some motivations are not clear.

---

> ### Author Response · Authors · 2021-11-21
> **Response to Reviewer DwFA**
>
>
> We sincerely thank the reviewer for appreciating our writing style and experimental findings. About the proposed concerns in three aspects, we will respond to them one by one as follows.
>
> $\textbf{Q1. [Concern on the novelty of the paper]}$
>
> (1) To the best of our knowledge, this is the first paper to address the noise edges problem when building a graph for GCNs on face images.
> Simultaneously, this paper demonstrates its causes, great impact, weaknesses in existing solutions and the challenges to solve it.
> We present our solutions corresponding to these challenges, which are logical and effective.
> This paper needs to be considered comprehensively rather than only focusing on the equations used.
>
>
> (2) To solve the noise edges problem mentioned above, several technical innovations are put forward.
> Four types of estimation methods for selecting neighbours are proposed and compared in Table 5, whose detailed processes are in Table 7. These methods cover the prediction of $Q$ and $\textbf{k}^{\text{off}}$ from the perspective of regression and classification.
> The proposed criterion of $Q$ evaluates the quality of candidate neighbours for each probe vertex. The extreme point of the $Q(j)$ curve is used to select neighbours.
> To our knowledge, this is the first paper to construct a graph from this perspective, although the criterion is motivated by the widely used F-score.
>
>
> (3) Several contributions are also made for the best performance of Ada-NETS.
> A variant version of Hinge loss in Equation 9 is used to control the distance between features of the same or different classes, aiming for a reasonable threshold for clustering.
> Huber loss in Equation 5 is adopted to handle the problem of imbalance distribution in the regression.
> The $\beta$ in $Q$ is also studied in original and structure space in Table 4.
> It is very difficult to reach such a remarkable performance by simply putting the proposed modules together.
>
>
> $\textbf{Q2. [Sample labels used in criterion}$ $Q$ $\textbf{]}$
>
> The graph building method is supervised and the clustering is unsupervised.
> Sample labels are used to calculate quality criterion $Q$ to train the  Adaptive Filter for building graphs in the training phase.
> In the testing phase, the trained Adaptive Filter is deployed to infer the extreme point of criterion $Q(j)$ curve on the unlabeled target dataset to construct graphs. With the built graphs, clustering with GCNs is conducted on them in an unsupervised manner.
>
>
> $\textbf{Q3. [Comparison with other graph optimization methods]}$
>
> Following the comments of the reviewer, we compare two more baseline graph optimization methods (e.g., the Threshold and  Chain Sim method).
> They are newly added in Figure 5(a).
> The detailed definitions of the 5 graph optimization methods are summarized in the newly added Table 10.
> Among the 5 methods, the proposed graph building in Ada-NETS achieves the best performance and shows its robustness to different $k$ settings (under different noise conditions).
>
> In addition, considering the ethical and privacy issues, all the datasets used in this paper are $\textbf{feature vectors}$ which can not restore to images.

---

### Decision · Program_Chairs · 2022-01-20

**Decision:**

Accept (Poster)

**Comment:**

All reviewers agree that the presented ADA-Nets approach is very interesting and sufficiently novel, addressing the degradation problem in face clustering. The reviewers are satisfied with the presented experimental studies in most cases. The rebuttal addressed a large majority of additionally raised questions. I disagree with one reviewer’s comment – that the focus of the paper is too narrow – because clustering techniques are of great interest to the ICLR community. I believe that the paper will be of interest to the audience attending ICLR and would recommend a presentation of the work as a poster.